# Numerical Simulation of Dynamic Mechanical Properties of Concrete under Uniaxial Compression

**DOI:** 10.3390/ma12040643

**Published:** 2019-02-20

**Authors:** Yijiang Peng, Qing Wang, Liping Ying, Mahmoud M. A. Kamel, Hongtao Peng

**Affiliations:** 1Key Laboratory of Urban Security and Disaster Engineering, Ministry of Education, Beijing University of Technology, Beijing 100124, China; pengyijiang@bjut.edu.cn (Y.P.); wangq@emails.bjut.edu.cn (Q.W.); mahmoud.kamel@fayoum.edu.eg (M.M.A.K.); 2Department of Civil Engineering, Faculty of Engineering, Fayoum University, 63514 Fayoum, Egypt; 3College of Water Resources and Civil Engineering, China Agricultural University, Beijing 100083, China; pwb@cau.edu.cn

**Keywords:** concrete, base force element method, strain-rate effect, meso-damage, dynamic behavior, numerical simulation

## Abstract

Based on the base force element method (BFEM), the dynamic mechanical behavior of concrete under uniaxial compression loading at different strain rates is investigated. The concrete can be considered as a three-phase composite material composed of aggregate, cement mortar, and interfacial transition zone (ITZ) on the meso-level. A two-dimensional random aggregate model is generated by the Monte Carlo method. A multi-linear two-dimensional damage model is applied to describe the damage properties of each phase in the concrete. The strain-softening behavior, strain-rate effect, and failure patterns of the concrete are studied. The numerical results find that the peaks of compressive stress and compressive strain of concrete show the rate-sensitivity in various degrees under different strain rates. The calculated results of the dynamic enhancement factors are in a good agreement with the formula given by the Comité Euro-International du Béton (CEB) and other experimental results. The failure diagram of the specimen clearly describes the compressive failure process of the concrete specimen. This failure’s characteristics are similar to the experimental results.

## 1. Introduction

In practice, the concrete used in infrastructure is usually subjected to dynamic loading, including impact loading and sustained loading. The previous obtained results show that the mechanical properties and damage characteristics of concrete under dynamic loading are very different from those under static loading [1,2]. Therefore, it is of interest to investigate the dynamic behavior of concrete under dynamic loading at high strain rates.

Hitherto, much research based on the traditional experimental test has been conducted. Abram [3] was the first to find out the compressive strength of concrete and show the strain rate sensitivity under dynamic loading. Bischoff et al. [4] concluded that the strain rate plays a significant role in both the dynamic ultimate strength and the dynamic deformation behavior of plain concrete at high strain rates. Ross et al. [5] and John et al. [6] respectively carried out Split-Hopkinson pressure bar (SHPB) tests to study the dynamic behavior of concrete under tensile stress. Malvar et al. [7] and Williams [8] respectively conducted literature reviews to investigate the strain-rate effects on the dynamic increase factor (DIF) of strength and elastic moduli; thus, the empirical equations were given correspondingly. In addition, the mechanical properties of the interfacial transition zone (ITZ) and the effect of the ITZ on the properties of rock-concrete materials were investigated [9]. Erzar et al. [10] developed direct tensile tests, spalling tests, and edge-on impact tests at high strain rates. The influence of the ITZ between the aggregates and the cement mortar on the uniaxial dynamic tensile strength of concrete was also studied. The SHPB apparatus was used to study the mechanical properties of cement-based materials. Deformation and stress distribution in the specimens are non-uniform due to the composite microstructure of the materials. The strength and failure patterns were also studied in both quasi-static and dynamic loadings [11,12,13,14,15,16,17]. Li et al. [18] used both cylinder and cube concrete specimens to conduct compressive tests and the effects of the specimen shape and size on concrete strength subjected to different loading rates were investigated. Jurowski and Grzeszczyk [19] concluded that the stabilized dynamic elastic modulus of concrete is proportional to the initial static elastic modulus, and the coefficient of proportionality is affected by the type of aggregates. Using various inner diameters of specimens, Zhang et al. [20] investigated the inertial effect on the tensile strength of concrete materials under dynamic loading.

In recent years, numerical analysis with computer modeling has become an efficient method to study the properties of concrete. On the basis of a damage parameter, Simo and Ju [21,22] proposed a continuum isotropic model and an anisotropic elastoplastic-damage model, and the results are consistent with the existing experimental data. The mechanical properties of small-eccentric loaded reinforced concrete (RC) columns and the dynamic behavior of steel fiber-reinforced concrete (SFRC) beams under impact loading were investigated using Abaqus software [23,24]. Furthermore, finite element theory is one of the major methods used to analyze the mechanical properties of cement-based composite material [25,26,27,28]. In the meso-scale, concrete can be deemed as a three-phase composite which comprises aggregate and cement mortar, with an ITZ between both. Regarding concrete studies, several scholars around the world have emphasized the relationship between microstructure and macroscopic mechanical properties of concrete under dynamic loading with a numerical method. 

Georgin et al. [29] exploited a viscous plastic model to simulate the SHPB test and studied the influence of inertial force and strain rates on concrete dynamic behavior. Snozzi et al. [30,31] and Gatuingt et al. [32,33] proposed a computational model to investigate the mechanical properties of concrete, composed of aggregate and mortar paste matrix, under dynamic loading of tension and compression. Also, Park et al. [34] analyzed the influence of impact loading at high strain rates on concrete-like materials using a dynamic finite element simulation. Zhou et al. [35,36] and Hao et al. [37,38] adopted numerical methods to analyze the influence of the ITZ on the dynamic failure patterns of the concrete, which is considered as a three-phased composite consisting of aggregate, mortar, and ITZ. Additionally, with the energy theory and the micro prestressed-solidification theory applied, Cusatis [39] utilized the developed Confinement Shear Lattice (CSL) model to analyze the effect of strain rates on the strength and failure behavior of concrete. However, Wu et al. [40] established a new experimental method of numerical simulation to identify the rate sensitive to the concrete dynamic tensile behavior. In this regard, in order to simulate the dynamic behavior of concrete under tensile, Zhou et al. [41] presented a two-dimensional meso-scale finite element model, validated by comparing with the experimental data from spall tests. Chen et al. [42] formulated a new dynamic compressive constitutive model applied to investigations of strain-rate effects and damage effects within the specimen. 

To date, the relationship between the failure mechanism of concrete and the stress distribution at high strain rate is not clear; however, the mesoscopic components remarkably affect the macroscopic mechanical properties of plain concrete. Therefore, on the meso-level, it is important to know the dynamic behavior of the concrete. 

In this paper, according to the investigations of concrete-like materials, under static loading with the base force element method (BFEM) proposed by Peng et al. [43,44,45], a dynamic base force element model is developed. In addition, the failure process of concrete under uniaxial dynamic loading at high strain rates is simulated. Strain-softening behavior and the failure process of modeled concrete under uniaxial compressive loading is also investigated. Finally, the effects of different loading velocities on compressive strength and the stress distribution in the specimens are studied.

## 2. Establishment of the Dynamic Base Force Element Equilibrium Equation

Generally, the behavior of real physical structures subjected to loadings or displacements is almost dynamic. According to the D’Alembert’s principle, considering the actions of inertial force and damping force, the balance equation of a single-degree-of-freedom system can be obtained as follows:(1)$$[M]\left\{\ddot{u}(t)\right\}+[C]\left\{\dot{u}\left(t\right)\right\}+[K_{d}]\left\{u\left(t\right)\right\}=\left\{P\left(t\right)\right\}$$
in which $\left\{\ddot{u}(t)\right\}$, $\left\{\dot{u}(t)\right\}$, and {u(t)} respectively indicate the acceleration vector quantity, the velocity vector quantity, and the displacement vector quantity of each node in structure; [M] means global mass matrix; [C] means the damping matrix; $\left[K_{d}\right]$ is the global stiffness matrix based on the base force as proposed in the paper; and {P(t)} is the dynamic load array of the structure.

Furthermore, the full variable form of the Newmark-β method is adopted to solve the dynamic equilibrium equation in the study.

### 2.1. Base Force Element Stiffness Matrix

According to the BFEM, which is based on the potential energy principle, a plane triangular element matrix $K^{IJ}$ [46], expressed by base force, can be obtained as follows (shown in Figure 1):(2)$$K^{IJ}=\frac{E}{2A(1+\nu )}\left[\frac{2\nu }{1-2\nu }m^{I}⊗m^{J}+m^{IJ}U+m^{J}⊗m^{I}\right]$$
where E, ν, and A mean the Young’s modulus, the Poisson’s ratio, and the area of an element, respectively; U is the unit tensor; $u^{I}$, $u^{J}$, $u^{K}$ represent the displacements of the triangular element vertex. 

For plane strain problems, *x* and *y* represent Cartesian coordinate system, and the element matrix $K^{IJ}$ can be expressed as follows:(3)$${\left[K_{IJ}\right]}^{e}=\frac{E}{2A(1+\nu )}\left[\begin{array}{cc} \frac{2-2\nu }{1-2\nu }m_{x}^{I}m_{x}^{J}+m_{y}^{I}m_{y}^{J} & \frac{2\nu }{1-2\nu }m_{x}^{I}m_{y}^{J}+m_{y}^{I}m_{x}^{J}\\ \frac{2\nu }{1-2\nu }m_{y}^{I}m_{x}^{J}+m_{x}^{I}m_{y}^{J} & \frac{2-2\nu }{1-2\nu }m_{y}^{I}m_{y}^{J}+m_{x}^{I}m_{x}^{J} \end{array}\right]$$
where $m^{IJ}=m^{I}\cdot m^{J}$, $m^{I}$ and $m^{J}$ can be described as follows and are shown in Figure 2:(4)$$m^{I}=m_{i}^{I}e_{i}=\frac{1}{2}(L_{IJ}n^{IJ}+L_{KI}n^{KI})=\frac{1}{2}(L_{IJ}n_{i}^{IJ}e_{i}+L_{KI}n_{i}^{KI}e_{i})=\frac{1}{2}(L_{IJ}n_{i}^{IJ}+L_{KI}n_{i}^{KI})e_{i},$$
(5)$$m^{J}=m_{i}^{J}e_{i}=\frac{1}{2}(L_{JK}n^{JK}+L_{IJ}n^{IJ})=\frac{1}{2}(L_{JK}n_{i}^{JK}e_{i}+L_{IJ}n_{i}^{IJ}e_{i})=\frac{1}{2}(L_{JK}n_{i}^{JK}+L_{IJ}n_{i}^{IJ})e_{i},$$
(6)$$\left\{\begin{array}{c} m_{x}^{I}\\ m_{y}^{I} \end{array}\right\}=\frac{1}{2}(L_{IJ}\left\{\begin{array}{c} n_{x}^{IJ}\\ n_{y}^{IJ} \end{array}\right\}+L_{LI}\left\{\begin{array}{c} n_{x}^{KI}\\ n_{y}^{KI} \end{array}\right\}),$$
(7)$$\left\{\begin{array}{c} m_{x}^{J}\\ m_{y}^{J} \end{array}\right\}=\frac{1}{2}(L_{JK}\left\{\begin{array}{c} n_{x}^{JK}\\ n_{y}^{JK} \end{array}\right\}+L_{IJ}\left\{\begin{array}{c} n_{x}^{IJ}\\ n_{y}^{IJ} \end{array}\right\}).$$
in which I, J, K represent the vertexes of the triangular element; $L_{IJ}$, $L_{KI}$, $L_{JK}$ are the lengths of the element boundary lines; $n^{IJ}$, $n^{KI}$, $n^{JK}$ are the normal vectors of the element boundary lines. Average strain components, which could take the place of the real strain in the case of small deformation, are given as follows: (8)$${\bar{\varepsilon }}_{x}=\frac{1}{A}\sum_{I=1}^{n}\left(u_{Ix}m_{x}^{I}\right),$$
(9)$${\bar{\varepsilon }}_{y}=\frac{1}{A}\sum_{I=1}^{n}\left(u_{Iy}m_{y}^{I}\right),$$
(10)$${\bar{\gamma }}_{xy}=\frac{1}{A}\sum_{I=1}^{n}\left(u_{Ix}m_{y}^{I}+u_{Iy}m_{x}^{I}\right).$$

### 2.2. Damping Matrix and Mass Matrix

In the finite element dynamic analysis, lumped mass matrix, one of the most common forms of the element mass matrix, is applied to simplify the calculation and reduce the storage space. According to the equivalent static principle, the mass of the element can be evenly distributed among three vertexes of a triangular element, according to the hypothesis that there is no interaction between the inertial forces of each vertex. 

In dynamic response problems it is generally assumed that the viscous damping force is proportional to the velocity of particle motion. 

Therefore, the effect of damping in the dynamic structural analysis at a high loading rate cannot be ignored. As a widely used orthogonal damping model, the Rayleigh damping model can be expressed as follows:(11)$$\left[C\right]=\alpha \left[M\right]+\beta \left[K_{d}\right],$$
(12)$$\alpha =\frac{2\zeta \omega_{1}\omega_{2}}{\omega_{1}+\omega_{2}},$$
(13)$$\beta =\frac{2\zeta }{\omega_{1}+\omega_{2}},$$
(14)$${\left[M\right]}^{e}=\frac{\rho_{b}TA}{3g}\;\left[\begin{array}{cccccc} 1 & 0 & 0 & 0 & 0 & 0\\ 0 & 1 & 0 & 0 & 0 & 0\\ 0 & 0 & 1 & 0 & 0 & 0\\ 0 & 0 & 0 & 1 & 0 & 0\\ 0 & 0 & 0 & 0 & 1 & 0\\ 0 & 0 & 0 & 0 & 0 & 1 \end{array}\right],$$
in which α and β are the mean coefficients of proportionality, which can be calculated from Equations (6) and (7); ζ means the damping ratio; $\omega_{1}$ and $\omega_{2}$ are the first and the second order angular frequencies of the specimen. 

It should be noted that, in the process of the meso-mechanical failure of concrete, there are few research works on damping models. The damping theory is a methodology which aims to describe the microscopic mechanism of damping in a macroscopic way. For the general structural analysis, Rayleigh damping is usually used to approximately describe the damping characteristics. Normally, the damping ratio of the engineering structure ranges from 0.01 to 0.1. The average value 0.05 is applied in this paper.

In this paper, to simplify the model, the influence of damage development on damping matrices is not considered. Consequently, the initial stiffness matrix of the specimen is adopted in the present dynamic analysis and the damping ratio is regarded as a constant. It is assumed that the mass matrix is independent from the damage state due to the conservation of mass.

## 3. Meso-Structure of Concrete

It is well known that concrete is a multi-phase heterogeneous brittle material. The mechanical properties of concrete are determined by components in its mixture. In this paper, concrete is described as a three-phase composite material at the meso-level, composed of coarse aggregates, cement mortar, and an ITZ between both phases. With the spherical aggregate applied, the microstructure in concrete can be depicted as in Figure 3. In the model only coarse aggregates with a particle size greater than 5 mm are represented clearly, while the other smaller aggregates are mixed up in the cement paste matrix. 

### 3.1. Random Aggregate Model

In this paper, in order to simulate the real distribution of aggregate as possible, the Monte Carlo method is used to establish the two-dimensional random aggregate model. Based on the Fuller three-dimensional aggregate gradation curve, Walraven [47] developed the two-dimensional cross-section aggregate gradation curve. The cumulative probability of aggregate particles satisfied the condition of $D<D_{0}$ (herein D means the diameter of the aggregate) is calculated using the following formula:(15)$$\begin{array}{cc} P_{c}\left(D-D_{0}\right)= & P_{k}[1.065{\left(\frac{D_{0}}{D_{\max}}\right)}^{1/2}-0.053{\left(\frac{D_{0}}{D_{\max}}\right)}^{4}-0.012{\left(\frac{D_{0}}{D_{\max}}\right)}^{6}\\ & -0.0045{\left(\frac{D_{0}}{D_{\max}}\right)}^{8}+0.0025{\left(\frac{D_{0}}{D_{\max}}\right)}^{10}] \end{array}$$
in which $P_{k}$ represents the percentage of the volume of aggregate in the whole concrete specimen; $P_{c}$ represents the cumulative probability of aggregate particles with a size smaller than $D_{0}$; $D_{\max}$ is the maximum diameter of the aggregate particles.

The number of aggregate particles meeting the requirement of $D_{1}<D<D_{2}$ can be calculated by the following equation: (16)$$n=\left[P_{c}\left(D<D_{2}\right)-P_{c}\left(D<D_{1}\right)\right]\times A/A_{i}$$
where A is the cross-section area of the concrete specimen and $A_{i}$ is the area of the representative particle diameter of the aggregate.

According to the theory of Fuller’s maximum density curve, three representative diameters (10 mm, 20 mm, and 32.5 mm) of aggregates are selected to calculate number of aggregate particles. The numbers are, respectively, 66, 11, and 3, as obtained from Equation (20). The aggregates calculated are put into the two-dimensional region, the same size as concrete specimen, by using the Monte Carlo method with three sets of different random numbers. Each aggregate should be in line with the boundary conditions and not overlap with the other aggregates before it is put on. The modeled aggregates are shown in Figure 4.

### 3.2. Mesh Generation Method and Element Attributes 

In the finite element analysis, two common methods are adopted to deal with the thickness of ITZ in order to simplify the calculation. One considers the thickness of the ITZ from 0.5 mm to 2 mm [48,49], and the other does not consider the thickness of the ITZ [50,51,52,53]. In this paper, the dynamic behavior of concrete is studied by using the former method and the size of the mesh element is deemed as the ITZ thickness to simplify the mesoscopic model.

The finite element mesh is shadowed on the random aggregate model. The different mechanical properties are assigned to the corresponding elements. A linear elastic triangular finite element grid is applied in the paper, shown in Figure 5. The type of element is determined by the position of element nodes. The element could be deemed as an aggregate (or cement mortar) element when the three nodes of the element are all located at the aggregate (or cement mortar) region; otherwise, the element is the ITZ when the element is located in the both the aggregate and cement mortar.

## 4. Dynamic Behavior for Concrete Meso-Components

### 4.1. Concrete Dynamic Damaged Model

As known, the mechanical properties of microscopic components have a great influence on the fracture damage behavior of concrete. Therefore, the heterogeneity of concrete should be taken into account in this numerical simulation. In this paper, concrete can be treated as a three-phase composite material composed of coarse aggregate, cement mortar, and the ITZ. Each phase material in concrete is assumed to be homogeneous and isotropic. The constitutive relation is presented in Figure 6. The maximum principal stress criterion is applied as the failure criteria in the study. The reduction of the elastic modulus E of the material in the damage processes can be expressed as follows:(17)$$E=E_{0}\left(1-D\right)$$
in which $E_{0}$ is the initial modulus of elasticity and D is the damage factor, defined as follows:(18)$$D_{t}=\{\begin{array}{cc} 0 & \varepsilon_{\max}\leq \varepsilon_{t0}\\ 1-\frac{\varepsilon_{t0}}{\varepsilon_{\max}}+\frac{\varepsilon_{\max}-\varepsilon_{t0}}{\eta_{t}\varepsilon_{t0}-\varepsilon_{t0}}\frac{\varepsilon_{t0}}{\varepsilon_{\max}}\left(1-\alpha \right) & \varepsilon_{t0}<\varepsilon_{\max}\leq \eta_{t}\varepsilon_{t0}\\ 1-\frac{\alpha }{\xi_{t}-\eta_{t}}\frac{\varepsilon_{\max}-\eta_{t}\varepsilon_{t0}}{\varepsilon_{\max}}+\frac{\alpha \varepsilon_{t0}}{\varepsilon_{\max}} & \eta_{t}\varepsilon_{t0}<\varepsilon_{\max}\leq \xi_{t}\varepsilon_{t0}\\ 1 & \varepsilon_{\max}>\xi_{t}\varepsilon_{t0} \end{array}$$
(19)$$D_{c}=\{\begin{array}{cc} 1-\frac{\beta }{\gamma } & \varepsilon_{\max}\leq \lambda \varepsilon_{c0}\\ 1-\frac{1-\beta }{1-\lambda }\frac{\varepsilon_{\max}-\lambda \varepsilon_{co}}{\varepsilon_{\max}}-\beta \frac{\varepsilon_{co}}{\varepsilon_{\max}} & \lambda \varepsilon_{c0}<\varepsilon_{\max}\leq \varepsilon_{c0}\\ 1-\frac{1-\gamma }{1-\eta_{c}}\frac{\varepsilon_{\max}-\varepsilon_{co}}{\varepsilon_{\max}}-\frac{\varepsilon_{co}}{\varepsilon_{\max}} & \varepsilon_{c0}<\varepsilon_{\max}\leq \eta_{c}\varepsilon_{c0}\\ 1-\frac{\gamma \varepsilon_{c0}}{\varepsilon_{\max}} & \eta_{c}\varepsilon_{c0}<\varepsilon_{\max}\leq \xi_{c}\varepsilon_{c0}\\ 1 & \varepsilon_{\max}>\xi_{c}\varepsilon_{c0} \end{array}$$
in which $\varepsilon_{0}$ means the principle strain; η denotes the residual strain coefficient; ξ represents the ultimate strain coefficient. 

In Figure 6, $f_{c}$ and $f_{t}$ stand for compressive and tensile strength; furthermore, the subscripts t and c respectively symbolize tensile and compressive features. The material parameters are given in Table 1, which is obtained after calculations.

### 4.2. Dynamic Increase Factor (DIF) for Concrete

In practice, damage patterns and mechanical properties of concrete under dynamic loading present different forms, which is called the strain-rate effect and is characterized by the dynamic increase factor (DIF). The dynamic increase factor for the compressive strength is recommended by Comité Euro-International du Béton [54] as the following:(20)$$DIF_{c}=f_{cd}/{f_{c}}^{\prime }=\left\{ \begin{array}{ll} {\left(\frac{\dot{\varepsilon }}{{\dot{\varepsilon }}_{s}}\right)}^{1.026\alpha } & \dot{\varepsilon }\leq 30s^{-1}\\ \gamma {\left(\frac{\dot{\varepsilon }}{{\dot{\varepsilon }}_{s}}\right)}^{1/3} & \dot{\varepsilon }>30s^{-1} \end{array}\right.$$
in which $f_{cd}$ is the dynamic uniaxial compressive strength and ${f_{c}}^{\prime }$ is the quasi-static uniaxial compressive strength; $\dot{\varepsilon }$ is the quasi-dynamic strain rate and ${\dot{\varepsilon }}_{s}$ is the quasi-static strain rate; $f_{cs}$ is the quasi-static uniaxial compression strength and $f_{c0}=10\;\mathrm{MPa}$. It should be noted that $\alpha =1/\left(5+9f_{cs}/f_{c0}\right)$ and lgγ=6.156α−2. As known from the empirical formula, the value of $DIF_{c}$ is 1 when the strain rate $\dot{\varepsilon }$ is $30\times 10^{-6}s^{-1}$, which is called the quasi-static load modal. In this paper, the minimum strain rate $10^{-3}s^{-1}$ is deemed as the quasi-static strain rate for comparison. According to the previous study, the strain-rate sensitivity of concrete characteristics, such as the Poisson’s ratio, elasticity modulus, energy dissipation capacity, and so forth, are much lower than the tensile and compressive strength of concrete [55].

## 5. Numerical Examples and Results

### 5.1. Boundary Conditions and Loading Model

In this section, the dynamic test of concrete subjected to uniaxial compressive loads is simulated with the BFEM. A standard concrete specimen, with the dimensions 150 mm × 150 mm, is chosen to conduct the test. The particle diameter of the coarse aggregates ranges from 5 mm to 40 mm. The boundary conditions are shown in Figure 7. The bottom and top surfaces of the specimen are restricted only in the *y*-direction; other surfaces of the specimen are free in all directions. The influence of friction between the specimen and the loaded end on the compressive strength of concrete is ignored.

Continuous and uniform vertical displacement loading is used on the specimens. The direction of vertical displacement loading is parallel to the *y*-axis. Dynamic displacement step-load is applied in this study and the velocity of displacement loading is well controlled by the duration of each load-step. Vertical strain rate ${\dot{\varepsilon }}_{m}$ under different loading velocity can be calculated as the following: (21)$${\dot{\varepsilon }}_{m}=\frac{2v}{h}$$
where v is the constant loading velocity and h is the height of the specimen. The loading curves of concrete specimens are shown in Figure 8. 

### 5.2. Dynamic Failure Behavior of Concrete under Uniaxial Compressive Stress 

Based on the meso-mechanical numerical simulation proposed in the present study, the specimen can be subdivided into triangular finite element meshes where the mesh size is 1 mm. With eight groups with different loading rates applied, dynamic uniaxial compression tests are carried out on the concrete specimens. The corresponding macroscopic nominal strain rates are $10^{-3}/s$, $10^{-2}/s$, $10^{-1}/s$, 1/s, 10/s, 30/s, 80/s, and 100/s. 

The stress–strain curves of concrete specimens are shown in Figure 9. The peak values of strain and stress are shown in Figure 10 at the different strain-rates. The damage process of concrete specimens under different strain-rate compression levels is shown in Figure 11. The failure patterns at different rates are illustrated in Figure 12. The distribution of the maximum principal stress in the damaged concrete is shown in Figure 13. It is to be noted that $\dot{\varepsilon }$ means the strain rate, ε is the strain, and σ is the compressive stress. In Figure 13, the positive numbers represent the elements under tensile stress and the negative numbers represent the elements are under compressive stress.

The stress–strain curves are similar to the static at low strain rates from 0.01/s to 0.1/s and the compressive strengths present a small enhancement. However, at the high strain rates from 1/s to 100/s, the curves slow down gradually as the strain rate increases and the stress peak and strain peak increase significantly, and especially the ratio of dynamic compressive strength at the rate of 100/s to the static compressive strength is about 2.67. It is also found that the increasing trend of the strain is more prominent than that of the stress from Figure 10. 

The meso-cracks first occur and spread in the ITZ, or the weak link, in cement mortar. Finally, a region is formed, filled with a concentrative zonal crack, which results in the destruction of concrete. At the high strain rates from 10/s to 100/s, the number of cracks greatly increases and the cracks are diffused through the whole concrete specimen—a few cracks even pass through the aggregate area. Coalescent cracks as a diffusion state can be seen in Figure 12. In addition, from Figure 9 and Figure 12, there is also a minor effect of the distribution of aggregates on the compressive stress. However, the distribution of aggregates can significantly affect the initiation and propagation of cracks.

In Figure 14, it can be observed that the curve of the present result goes up precipitously at the strain rate of 0.1/s. The results obtained show good agreement with the Comité Euro-International du Béton (CEB) standard and other experimental data at the strain rates from 10/s to 100/s. At the strain rates from 0.01/s to 10/s, the DIF is lower than the CEB standard.

## 6. Discussion

The BFEM for dynamic analysis of concrete introduced in the present study verifies an efficient numerical simulation method to investigate the damage mechanism of concrete. The strain-rate effect and the failure characteristics are explored using two-dimensional models. Several conclusions can be obtained as follows:

(1) The present results prove that the failure process of concrete under dynamic compression is simulated well. With the increase of stress, the material enters the nonlinear stage and the stress–strain curves show a nonlinear increase relationship, simultaneously.

(2) The variation tendency of the DIF of concrete at different strain rates is consistent with the available experimental test data and the CEB empirical formula.

(3) At high strain rates, cracks increase and, in a diffusion state, some of the elements present fracture damage and more energy is released, which could enhance the dynamic strength of the concrete.

(4) The strain peaks of concrete present rate-sensitivity under different strain rates similar to stress peaks.

For future work, a three-dimensional model of dynamic problems could be addressed by the BFEM and other conditions could be considered; for instance, considering the shape of aggregates, complicated boundary constraints, and so on, so as to simulate real concrete as much as possible.

## Figures and Tables

**Figure 1 materials-12-00643-f001:**
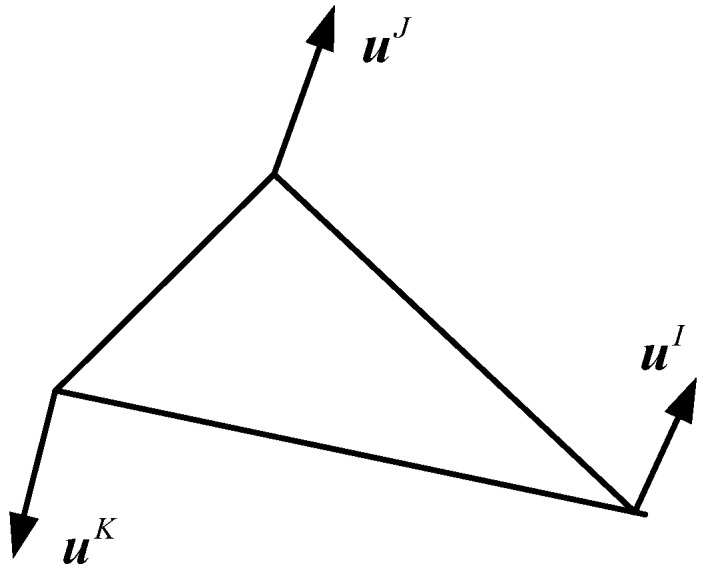
A triangular element.

**Figure 2 materials-12-00643-f002:**
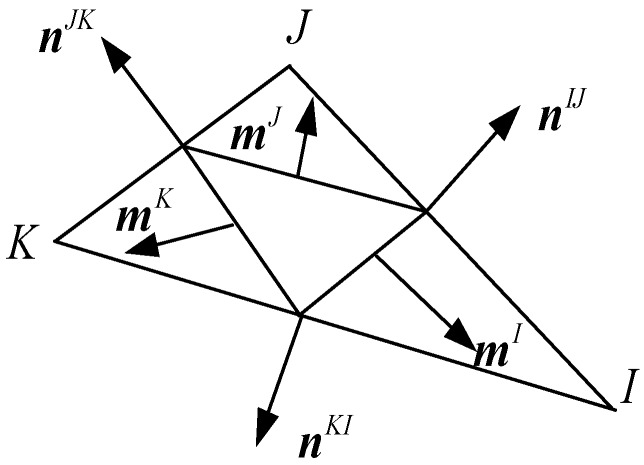
Construction of a triangular element stiffness.

**Figure 3 materials-12-00643-f003:**
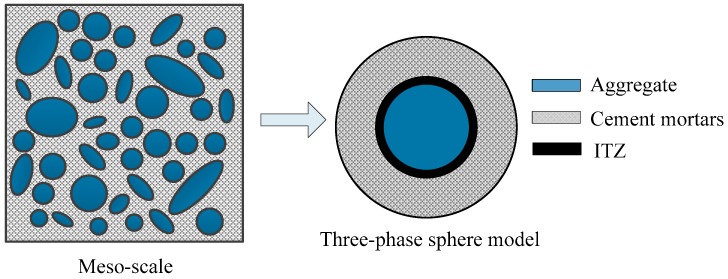
Three-phase sphere model of concrete.

**Figure 4 materials-12-00643-f004:**

Random aggregate models of concrete. (**a**) Specimen 1; (**b**) specimen 2; (**c**) specimen 3.

**Figure 5 materials-12-00643-f005:**
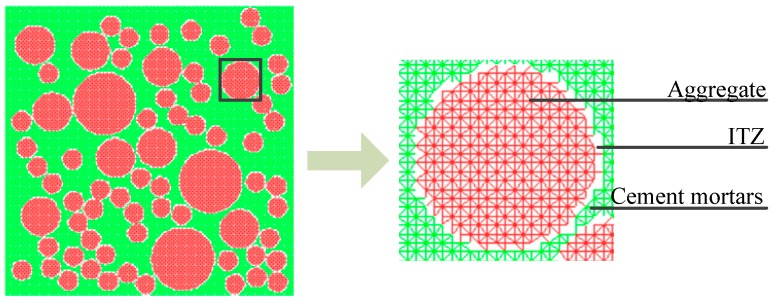
The diagram of mesh generation.

**Figure 6 materials-12-00643-f006:**
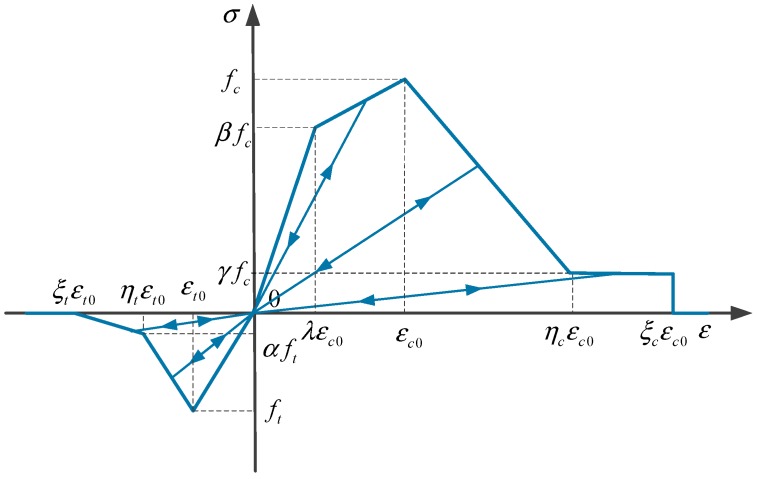
Mechanical constitutive model of materials.

**Figure 7 materials-12-00643-f007:**
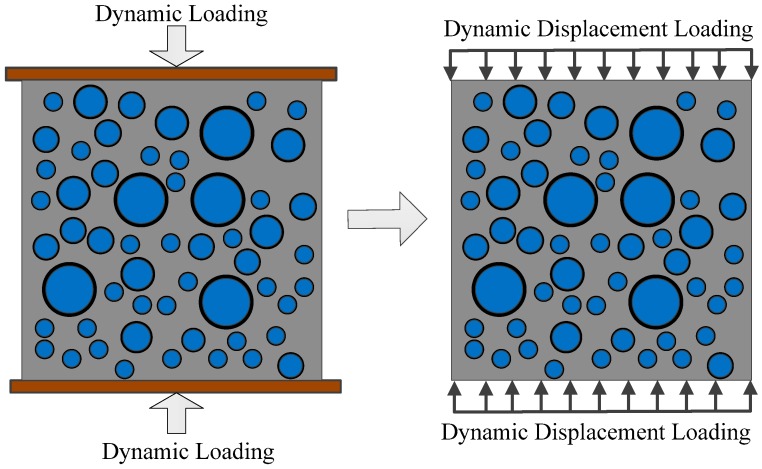
Loading model of compression test.

**Figure 8 materials-12-00643-f008:**
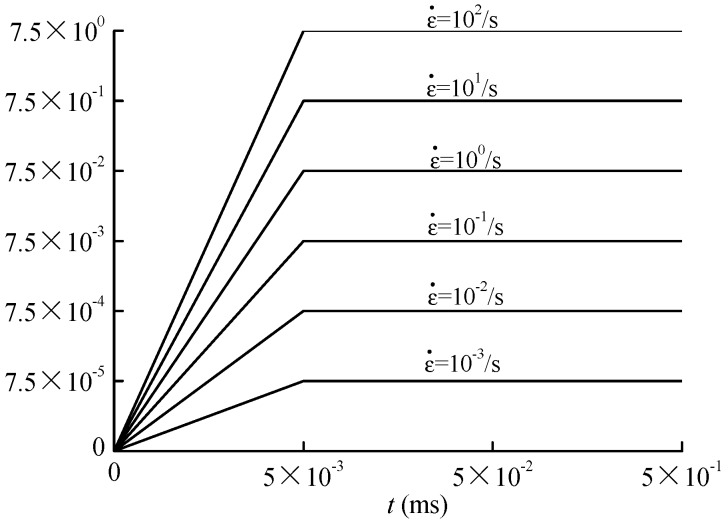
Loading curve of dynamic uniaxial compression test.

**Figure 9 materials-12-00643-f009:**
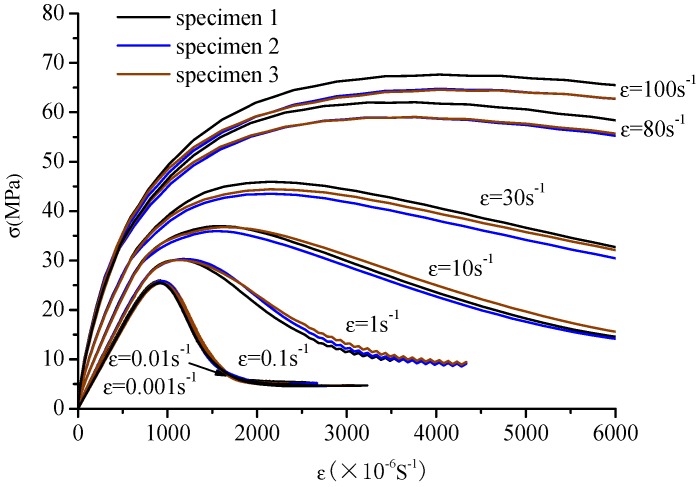
Stress–strain curve of concrete under dynamic uniaxial compressive stress at different strain rates.

**Figure 10 materials-12-00643-f010:**
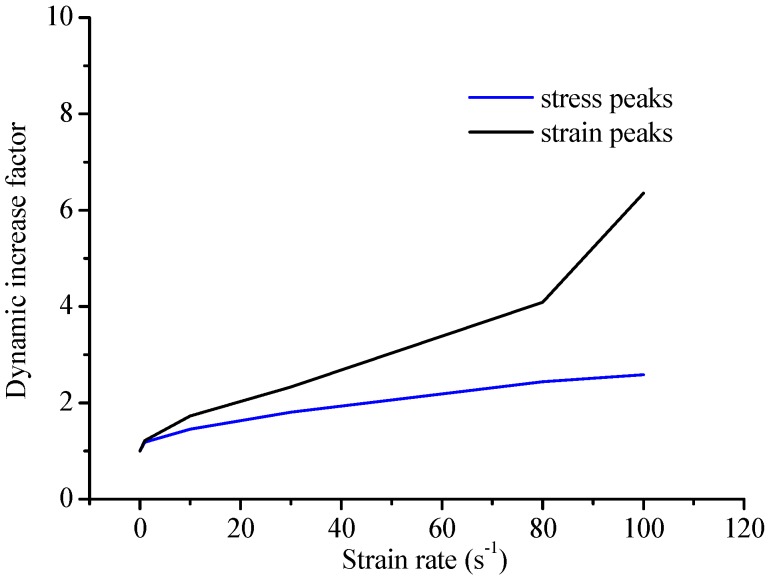
The comparison of the peak values of strain and stress.

**Figure 11 materials-12-00643-f011:**
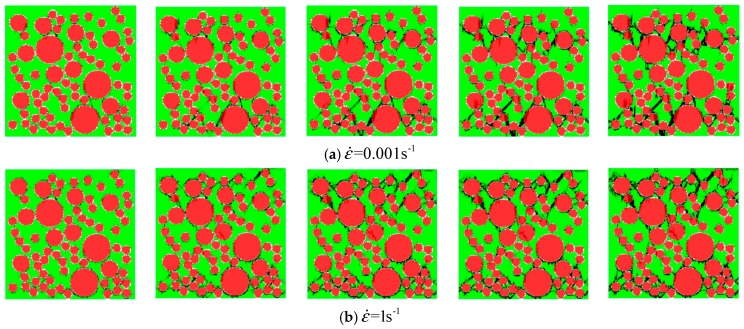
The damage process of concrete specimen 1 under dynamic uniaxial compressive stress; (**a**) $\dot{\varepsilon }=0.001\;s^{-1}$; (**b**) $\dot{\varepsilon }=1\;s^{-1}$; (**c**) $\dot{\varepsilon }=10\;s^{-1}$; (**d**) $\dot{\varepsilon }=100\;s^{-1}$.

**Figure 12 materials-12-00643-f012:**
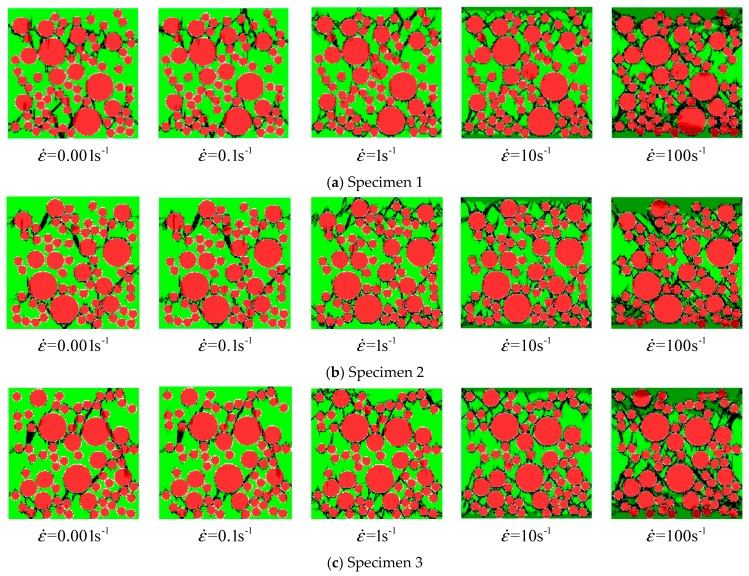
The failure pattern of concrete specimens under dynamic uniaxial compressive stress; (**a**) specimen 1; (**b**) specimen 2; (**c**) specimen 3.

**Figure 13 materials-12-00643-f013:**
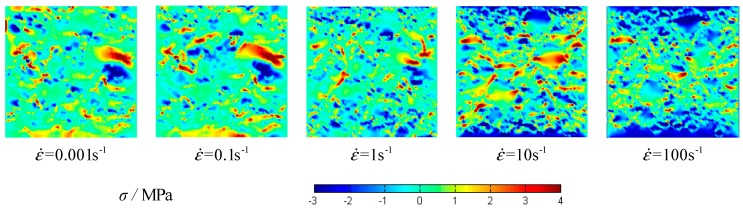
The distribution of maximum principal stress at different strain rates of specimen 1.

**Figure 14 materials-12-00643-f014:**
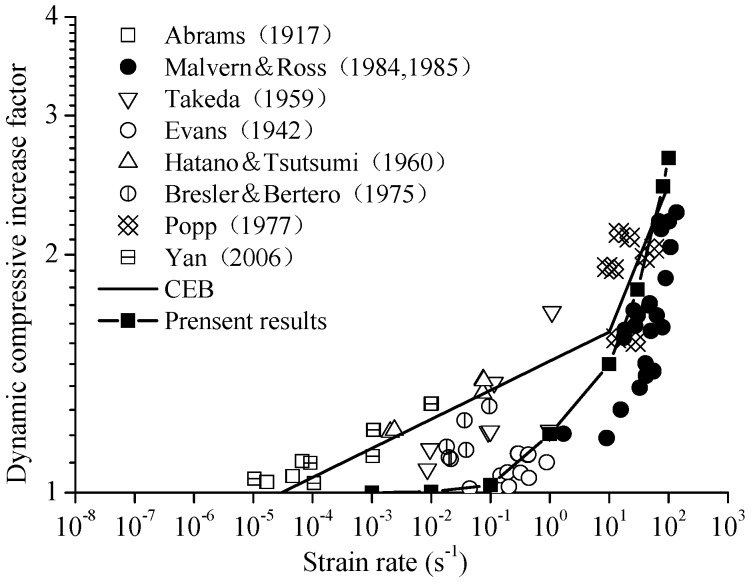
The effect of strain rate on compressive strength of plain concrete.

**Table 1 materials-12-00643-t001:** Mechanical parameters of materials.

Mechanical Parameters	Cement Mortar	Interfacial Transition Zone (ITZ)	Aggregate
Density ρ (kg/m^3^)	2100	1700	2700
Poisson’s ratio ν	0.22	0.2	0.16
Strength (tensile/compressive) σ (MPa)	3.2/32	2.5/25	7/70
λ	0.25	0.25	0.80
β	0.85	0.65	0.90
γ	0.35	0.35	0.35
α	0.3	0.3	0.3
$\eta_{t}/\eta_{c}$	4/4	3/3	5/5
$\xi_{t}/\xi_{c}$	10/10	10/10	10/10

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
