# Peer review of "Numerical Simulation of Dynamic Mechanical Properties of Concrete under Uniaxial Compression"

_materials, 2019, doi:10.3390/ma12040643_

Round 1
Reviewer 1 Report
General comments
The paper delas with the application of the base force element method (BFEM) to study the dynamic mechanical behavior of 9concrete under uniaxial compression loadings at different strain rates.
The failure process of concrete under uniaxial dynamic loadings at high strain-rates is simulated. Strain-softening behavior and the failure process of modeled concrete under uniaxial compressive load are also investigated, and the effect of different loading velocity on compressive strength and the stress distribution in the specimens are studied.
Interestin agreement with published experimental data a re presented.
Remarks
Line 109 Please refer to Fig. 1.
Line 116 use "Fig. 2" instead of "Fig.2" also in other parts of the paper.
Line 146 Please add in the text considerations about the value of the damping ratio = 0.05 assumed.
Line 158 please use a space between number and units "5 mm" instead of "5mm", also in other parts of the paper
Line 176 please explain in the paper how the aggregate particles are distribuite in the 2D domain whem the result of equation 820) are available
Line 181 Please provide in paper detail about he size of the ITZ layer and the criteria to establish this size.
Section 5.1 include details about the pplied load, but explanation of the applied constraint, shown in Fig. 6 (right) are missing. Pleaseinclude also a discussion about the agreement between this ideal constraint and the constraint of the actual experimental tests.
Line 251 Fig.12, the positive numbers represent tensile stress and the negative numbers represent compressive stress.
But Line 210 The dynamic increase factor (DIF) for the compressive strength is recommended
by Comité Euro-International du Béton [48] , what about the traction strenght?
Author Response
Response to Reviewer 1 Comments
Dear Reviewer 1:
Thank you for receiving your reply about the manuscript entitled “Numerical Simulation on Dynamic Meso-damage Analysis of Concrete under Uniaxial Compressive Stress” (ID:materials-446004). Those comments are all valuable and very helpful for revising and improving our paper, as well as the important guiding significance to our researches. We have studied comments carefully and have made correction which we hope meet with approval. Revised portion are marked in blue in the paper. The main corrections in the paper and the responds to the reviewer’s comments are as flowing:
(1) After careful deliberation, we decide to entitle “Numerical Simulation of Dynamic Mechanical Properties of Concrete under Uniaxial Compression” instead of the original title.
(2) The manuscript has been checked by Mahmoud Mohamed Ali Kamel a native English speaking colleague, the paper has been revised in terms of English and sentence structure. We also add Mahmoud Mohamed Ali Kamel as a co-author.
(3) With the help of Hongtao Peng, the test data and the result in the study have been dealt well. Therefore, we add Hongtao Peng as a co-author in the paper.
Point 1: Line 109 Please refer to Fig. 1.
Response 1: The paper line 110 has referred to Fig. 1.
Point 2: Line 116 use "Fig. 2" instead of "Fig.2" also in other parts of the paper.
Response 2: The format has been modified in full text.
Point 3: Line 146 Please add in the text considerations about the value of the damping ratio = 0.05 assumed.
Response 3: Normally, the damping ratio of the engineering structure range from 0.01 to 0.1. It should be noted that, in the process of meso-mechanical failure of concrete, there are few researches on damping models. The damping theory is a methodology which is aimed to describe the microscopic mechanism of damping in a macroscopic way. For the general structural analysis, Rayleigh damping is usually used to approximately describe the damping characteristics in an average sense. Considering the above factors, the average value 0.05 is applied in the paper, and the text considerations about the value of the damping ratio have been added in the section 2.2.
.
Point 4: Line 158 please use a space between number and units "5 mm" instead of "5mm", also in other parts of the paper.
Response 4: The format has been modified in full text.
Point 5: Line 176 please explain in the paper how the aggregate particles are distributed in the 2D domain when the result of equation (20) are available
Response 5: According to the theory of Fuller's maximum density curve, three representative diameter (10 mm, 20 mm and 32.5 mm) of aggregates are select to calculate number of aggregate particle, and the numbers are respectively 66, 11 and 3 obtained from the equation 20. The aggregates calculated are put into the two dimensional region by using Monte Carlo method. Every aggregate should be in line with the boundary conditions and not overlapped with the other aggregates before it is put on. The result show that the minor effect of distribution of aggregates on compressive stress, however, the distribution of aggregates can be significantly affect the initiation and propagation of cracks. Three random aggregate models are generated shown in the section 3.1 and simulated in the section 5.2. The Fig. 9 and the Fig. 12 are also revised in the paper.
Point 6: Line 181 Please provide in paper detail about the size of the ITZ layer and the criteria to establish this size.
Response 6: The finite element mesh is shadowed on the random aggregate model. The different mechanical properties are assigned to the corresponding elements according to the position of element node: the element could be deemed as aggregate (or cement mortar) element when the three nodes of the element are all located at the aggregate (or cement mortar) region; otherwise, the element is the ITZ when the element is located in the both aggregate and cement mortar. Lots of work has illustrated the mechanical properties of the ITZ, especially the dynamic behaviour, are not well explained yet. Therefore, in the finite element analysis, two common methods are adopted to deal with the thickness of ITZ for simplifying the calculation. One is considering the thickness of the ITZ from 0.5 mm to 2 mm (Du et al., 2013; Jayasuriya et al., 2018;), and the other is without considering the thickness of the ITZ (Nilsen et al.,1993; Agioutantis et al., 2002; Fakhari et al., 2013; Du et al., 2014). In this paper, the dynamic behaviour of concrete is studied by using the former, and the size of the mesh element is deemed as the ITZ thickness to simplify the mesoscopic model. The revision is shown in the section 3.2 of the paper.
Point 7: Section 5.1 include details about the applied load, but explanation of the applied constraint, shown in Fig. 6 (right) are missing. Please include also a discussion about the agreement between this ideal constraint and the constraint of the actual experimental tests.
Response 7: The boundary conditions have been added in the paper: the bottom and top surfaces of the specimen are restricted only in the y- direction; other surfaces of the specimen are free in the all directions. In an actual experimental test, in order to reach the ideal constraint state, some measures would be taken to reduce the friction between the specimen and the loaded end, such as using some kind of lubricant in the surface of specimen. Therefore, the influence of friction between the specimen and the loaded end on the compressive strength of concrete is ignored in the study to simplify the calculation. The revision is shown in the section 5.1 of the paper.
Point 8: Line 251 Fig.12, the positive numbers represent tensile stress and the negative numbers represent compressive stress. But Line 210 The dynamic increase factor (DIF) for the compressive strength is recommended by Comité Euro-International du Béton [48] , what about the traction strength?
Response 8: The text is mainly simulating the compressive test on concrete under the dynamic loading. The dynamic enhancement factors of the compressive strain and stress of concrete are investigated. The Fig.12 (now Fig.13) describes the distribution of each element maximum principal stress in the interior of specimen. The positive numbers represent the elements are under tensile and the negative numbers represent the element are under compression. Therefore, the traction strength is not mentioned. The revision is shown in line 284 to 285 of the paper.
Special thanks to you for your good comments.
We tried our best to improve the manuscript and made some changes in the manuscript. These changes will not influence the content and framework of the paper. And here we did not list the changes but marked in blue in revised paper.
We appreciate for your warm work earnestly, and hope that the correction will meet with approval.
Once again, thank you very much for your comments and suggestions.
Yours sincerely

Reviewer 2 Report
The paper is interesting, well-conceived and well structured. The reviewer, would recommend the authors to thoroughly revise the paper in terms of English and sentence structure to make the paper more accessible to a wider audience.
Author Response
Response to Reviewer 2 Comments
Dear Reviewer 2:
Thank you for receiving your reply about the manuscript entitled “Numerical Simulation on Dynamic Meso-damage Analysis of Concrete under Uniaxial Compressive Stress” (ID:materials-446004). Those comments are all valuable and very helpful for revising and improving our paper, as well as the important guiding significance to our researches. We have studied comments carefully and have made correction which we hope meet with approval. Revised portion are marked in blue in the paper. The main corrections in the paper and the responds to the reviewer’s comments are as flowing:
(1) After careful deliberation, we decide to entitle “Numerical Simulation of Dynamic Mechanical Properties of Concrete under Uniaxial Compression” instead of the original title.
(2) The manuscript has been checked by Mahmoud Mohamed Ali Kamel a native English speaking colleague, the paper has been revised in terms of English and sentence structure. We also add Mahmoud Mohamed Ali Kamel as a co-author.
(3) With the help of Hongtao Peng, the test data and the result in the study have been dealt well. Therefore, we add Hongtao Peng as a co-author in the paper.
Point 1: The paper is interesting, well-conceived and well structured. The reviewer, would recommend the authors to thoroughly revise the paper in terms of English and sentence structure to make the paper more accessible to a wider audience
Response 1: The manuscript has been checked by Mahmoud Mohamed Ali Kamel a native English speaking colleague, the paper has been revised in terms of English and sentence structure, hoping that can make more accessible to a wider audience. Thanking you for your suggestions and your time on my paper.
Special thanks to you for your good comments.
We tried our best to improve the manuscript and made some changes in the manuscript. These changes will not influence the content and framework of the paper. And here we did not list the changes but marked in blue in revised paper.
We appreciate for your warm work earnestly, and hope that the correction will meet with approval.
Once again, thank you very much for your comments and suggestions.
Yours sincerely

Round 2
Reviewer 2 Report
The paper was revised.